# Dose–Response Relationships between Levels of Alcohol Use and Risks of Mortality or Disease, for All People, by Age, Sex, and Specific Risk Factors

**DOI:** 10.3390/nu13082652

**Published:** 2021-07-30

**Authors:** Jürgen Rehm, Pol Rovira, Laura Llamosas-Falcón, Kevin D. Shield

**Affiliations:** 1Centre for Addiction and Mental Health, Institute for Mental Health Policy Research, 33 Ursula Franklin Street, Toronto, ON M5S 2S1, Canada; llamosasfalcon@gmail.com (L.L.-F.); Kevin.Shield@camh.ca (K.D.S.); 2Institute of Clinical Psychology and Psychotherapy, Technische Universität Dresden, Chemnitzer Str. 46, 01187 Dresden, Germany; 3Center for Interdisciplinary Addiction Research (ZIS), Department of Psychiatry and Psychotherapy, University Medical Center Hamburg-Eppendorf (UKE), Martinistraße 52, 20246 Hamburg, Germany; 4Dalla Lana School of Public Health, University of Toronto, 155 College Street, Toronto, ON M5T 1P8, Canada; 5Faculty of Medicine, Institute of Medical Science, University of Toronto, Medical Sciences Building, 1 King’s College Circle, Room 2374, Toronto, ON M5S 1A8, Canada; 6Centre for Addiction and Mental Health, Campbell Family Mental Health Research Institute, 33 Ursula Franklin Street, Toronto, ON M5S 3M1, Canada; 7Department of Psychiatry, University of Toronto, 250 College Street, 8th Floor, Toronto, ON M5T 1R8, Canada; 8Department of International Health Projects, Institute for Leadership and Health Management, I.M. Sechenov First Moscow State Medical University (Sechenov University), Trubetskaya Street 8, b. 2, 119991 Moscow, Russia; 9Program on Substance Abuse, Public Health Agency of Catalonia, 08005 Barcelona, Spain; polrovira26@gmail.com; 10Hospital Universitario 12 de Octubre, Av. de Córdoba, s/n, 28041 Madrid, Spain

**Keywords:** alcohol, patterns of drinking, disease, mortality, dose response, monotonous, protective effects, curvilinear, alcohol control policy

## Abstract

Alcohol use has been causally linked to more than 200 disease and injury conditions, as defined by three-digit ICD-10 codes. The understanding of how alcohol use is related to these conditions is essential to public health and policy research. Accordingly, this study presents a narrative review of different dose–response relationships for alcohol use. Relative-risk (RR) functions were obtained from various comparative risk assessments. Two main dimensions of alcohol consumption are used to assess disease and injury risk: (1) volume of consumption, and (2) patterns of drinking, operationalized via frequency of heavy drinking occasions. Lifetime abstention was used as the reference group. Most dose–response relationships between alcohol and outcomes are monotonic, but for diabetes type 2 and ischemic diseases, there are indications of a curvilinear relationship, where light to moderate drinking is associated with lower risk compared with not drinking (i.e., RR < 1). In general, women experience a greater increase in RR per gram of alcohol consumed than men. The RR per gram of alcohol consumed was lower for people of older ages. RRs indicated that alcohol use may interact synergistically with other risk factors, in particular with socioeconomic status and other behavioural risk factors, such as smoking, obesity, or physical inactivity. The literature on the impact of genetic constitution on dose–response curves is underdeveloped, but certain genetic variants are linked to an increased RR per gram of alcohol consumed for some diseases. When developing alcohol policy measures, including low-risk drinking guidelines, dose–response relationships must be taken into consideration.

## 1. Introduction 

Alcohol use has been causally linked to more than 200 disease and injury conditions (based on three-digit ICD-10 codes; [1,2] and Table 1 below), indicating that alcohol use alone is a necessary cause or that it is a component cause (for an epidemiological definition of causality see: [3]). Different dimensions of alcohol use causally lead to a modified risk of disease and injury [4]. Average levels of drinking over time have been linked to many chronic disease categories, such as cancer, gastrointestinal disease, and different categories of heart disease [2]. Heavy drinking occasions are more linked to the more immediate effects of alcohol use, such as unintentional and intentional injuries [5,6], but also affect the risk for ischemic diseases [7], and infectious diseases (e.g., HIV, [2]). Accordingly, these two dimensions of alcohol use are most often used in epidemiological studies to predict disease and injury risk (the average level of consumption over time, and heavy episodic drinking (HED)).

The dimensions of average volume of alcohol use and HED are not independent: all people with chronic heavy drinking, such as many of those with alcohol use disorders (but see [9,10]), engage in HED [11]. People who do not engage in chronic heavy drinking can be separated in two categories: those who engage in HED and those who do not, with different impacts on some disease and mortality outcomes [12]. For example, on average, light drinking has been associated with cardio-protectivity compared to lifetime abstention [13]; these effects disappear for light drinkers who also engage in HED [7]. Furthermore, the risk of unintentional injury depends on the blood alcohol concentration and thus is highest in people who have engaged in a heavy drinking occasion prior to sustaining their injury [5,6]. However, this dose–response relationship has been shown to vary based on prior drinking experience. Thus, Gmel and colleagues [14] showed that while people at all average drinking levels are at increased risk for alcohol-related injury, those who normally drink lightly are at higher risk of injury compared to chronic heavy drinkers after consuming the same quantity of alcohol. Similar results have been reported in other studies [15].

It is important to know the exact dose–response relationships between alcohol use and disease outcomes because alcohol control policy measures will in part depend on these relationships [16]. For example, if most dose–response relationships are linear, population measures to lower the population mean of alcohol use (=per capita consumption), such as taxation increases or availability restrictions, are the most effective and appropriate measures (e.g., [17]). If the dose–response relationships are steep and exponential, measures for heavy drinkers could potentially be the more effective and/or cost-effective strategy [16,18]. We will come back to these choices in the Discussion.

Given the numerous diseases which are causally related to alcohol consumption (see Table 1 above), this contribution will systematically examine dose–response curves between different average levels of alcohol use and disease outcomes with a focus on modification of such curves by personal characteristics and/or the drinking context. Thus, this contribution will explore the impact of heavy episodic drinking on dose–response curves, as well as if these risks differ by factors such as sex, age, socio-economic status, genetic constitution, and behavioural risk factors.

## 2. Materials and Methods

This paper is a narrative review of the relative risk (RR) functions between average level of alcohol use and the occurrence of diseases and injuries [19], mainly based on meta-analytically derived dose–response curves used for comparative risk assessments (e.g., [8,20,21]). We used the meta-analyses reported by the latest WHO comparative risk assessments and from the Global Burden of Disease Study as they are comprehensive, evaluated by special committees, and continuously updated.

These dose–response curves usually compare the relative change in risk from a certain level of average drinking against the risk of a lifetime abstainer (for the rationale, see [22]). Lifetime abstention is selected as a comparison group, rather than abstention, as the people constituting the latter group are comprised of lifetime abstainers and former drinkers, and therefore have different risk levels (for further discussion, see [23,24]). In burden calculations, the theoretical minimum risk exposure level for comparison, rather than abstention, has traditionally been used (e.g., [21]) but, for the topic of this paper, this is not relevant. Whenever possible, the relative-risk curves are sex-specific ([8,20]; for further discussion, see below).

In risk factor epidemiology, in particular in comparative risk assessments, these dose–response curves based on RR for different exposures are then applied to almost all countries, taking into account the respective population distributions for drinking [25]. The only exception is the Russian Federation and surrounding countries, where region-specific dose–response curves are usually applied [26], because the same average level of drinking has been found to be associated with higher risks of mortality and other harms [27]. This procedure assumes that the dose–response curve is fairly stable, an assumption which will be examined below.

## 3. Results

### 3.1. Basic Typologies of Dose–Response Relationships

*Threshold effects*: Most risk curves can be described as continuous, but there is some evidence for threshold effects related to alcohol use. Tuberculosis (TB) provides us with an excellent example of this. In the original meta-analysis, Lönnroth and colleagues [28] examined the relationship between different levels of alcohol consumption, and concluded that all studies with alcohol use below a threshold of 40 g pure alcohol per day found no significant relationship with incidence of active TB, whereas drinking above this threshold resulted in about a three-fold higher risk. In this study, people with alcohol problems, including use disorders, were classified as being above the threshold. A newer study on the alcohol-TB dose–response relationship corroborated these results for people with alcohol problems, but did not identify a threshold when they included only studies involving individuals with an average volume of alcohol consumption [29]. In these studies, a dose–response relationship was found, which can be explained by a monotonic increasing risk. In the linear continuous meta-analysis which was identified as the best-fitting model, the TB risk rose by about 2% per gram pure alcohol intake (95% CI: 0–3%), leading to the following RR: at 25 g/day: 1.57 (95% CI: 1.10–2.23), at 50 g/day: 2.46 (95% CI: 1.21–4.98), at 75 g/day: 3.85 (95% CI: 1.33–11.11), and at 100 g/day: 6.03 (95% CI: 1.47–24.81).

In the current comparative risk analyses [8], there is only one threshold relationship for the risk of HIV/AIDS. This model is not based on meta-analyses but on experimental data, and the threshold of 48 g/day and 60 g/day is modelled based on reaching a minimal blood alcohol level every day [2]. In sum, there is no good evidence for a real threshold effect, but for one dose–response relationship, a conservative threshold was chosen. It was recently decided that the risk for sexual transmitted diseases other than HIV/AIDS would be modelled in the same manner in the next Global Status Report on Alcohol and Health (based on [30]).

*Monotonic dose–response relationships*: Most dose–response relationships are monotonic [11] if the comparisons are made with lifetime abstainers rather than with the combined group of lifetime abstainers and former drinkers. This means that with increasing average alcohol consumption, the risk for disease or mortality increases. In some instances, a monotonic relationship means that the risk between levels of alcohol use and RR is best modelled linearly, while in others there are exponential or flattening functions [11]. It should be noted that as the RR functions are exponential, an exponential linear function means that the underlying disease or mortality risk is exponential.

Exceptions seem to be ischemic disease and diabetes, which show curvilinear relationships, with light to moderate drinkers showing less risk than lifetime abstainers (for details, see below).

*Curvilinear dose–response relationships*: A number of important disease and mortality outcomes seem to show curvilinear relationships with the lowest risk at low to moderate drinking levels: ischemic heart disease [7,13], ischemic stroke [31]; diabetes [32,33]. As for ischemic disease, the protective effect seems to be mainly for acute outcomes, especially acute myocardial infarction, and less so for chronic ischemic events [34]. The use of these curvilinear dose–response relationships has received criticism, mainly because of their underlying unclear comparison groups, resulting in either overestimating the protective impact of alcohol or in artificially creating such an impact where there is in reality a monotonous relationship (e.g., [35]). However, at least for ischemic disease categories, there are plausible biological pathways for a protective effect from light to moderate drinking [36,37], so the overall shape of the curve is likely curvilinear with some kind of protective effect before the curve rises up again. However, the protective effect is likely overestimated, especially if meta-analyses based on overall abstention is used; the more former drinkers included, the higher the overestimation (for ischemic heart disease, a meta-analysis estimated the RR of former drinkers for ischemic heart disease mortality; 1.25, 95% CI: 1.15–1.36; and 1.54, 95% CI: 1.17–2.03 for men and women, respectively [38]).

Another controversial dose–response curve involves alcohol use and dementia. Most reviews found a curvilinear relationship with a protective effect for light to moderate drinking [39], even though heavy drinking is clearly detrimentally related to the incidence of dementia, in particular early onset dementia [40].

A final consideration here concerns sex differences in curvilinear relationships: the protective effect and the increase in risk after the nadir are more pronounced in women, both for ischemic disease and diabetes (see risk curves in Appendix 1 of [8]). For more on sex as a modifier, see Section 3.2.

### 3.2. Modifiers of Dose–Response Relationships

*Sex*: In all countries, men, on average, consume higher quantities of alcohol, and have more heavy drinking occasions [41]. Accordingly, alcohol-attributable mortality or burden of disease rates are higher in men [8,21]. However, the differences in health harms are attenuated somewhat, as many RRs, especially for chronic diseases, are lower for men for the same level of drinking. In the classical comparative risk assessment of English and colleagues [42], a categorical approach was used with categories of <20 g/day, 20–40 g/day, and >40 g/day for women, and <40 g/day, 40–60 g/day and >60g/day for men, and the RRs were often estimated to be similar for the respective categories. Thus, in the analysis, the RR for the first category of women with a midpoint of 10 g/day was the same as for the first category of men with a midpoint of 30 g/day.

These early quantifications were corroborated by later meta-analyses, and for the following conditions, higher RRs are currently seen for women: HIV, hypertensive heart disease, ischemic heart disease, both stroke types, liver cirrhosis, and pancreatitis. For instance, Figure 1 is based on the most recent comprehensive meta-analysis of Roerecke and colleagues [43] and shows a much higher RR in women for liver cirrhosis at the same level of drinking.

It should be noted that the categorical analyses based on a larger number of studies resulted in fewer exponential dose–response curves, but the women had higher risks for the same amount of drinking. For instance, a consumption of 7 and more drinks per day in women results in a RR of 24.6 (95% CI: 14.8–40.9), whereas the same amount of average daily drinking in men was associated with a RR of 6.9 (95% CI: 1.1–45.0). These differences are based on the fact that for dose–response relationships in the cohorts used in medical epidemiology, there is often not a sufficient number of people included to estimate dose–response curves for higher levels of average drinking, such as those seen in some treatment samples (see Discussion below and [44]).

Before discussing other modifiers, it should be noted that, even now, more than 25 years after the first comparative risk assessment, there is still often not a sufficient number of studies available to adequately separate risk curves between the sexes (see [8,20] for an overview). Data scarcity is even more of a problem for other modifiers (see below and the Discussion).

*Age*: There are biological and other reasons that dose–response curves for alcohol use should change with age. However, the underlying literature is scarce. Ischemic disease categories are an exception. Klatsky found an attenuation of risk based on age [45]. Based on this, the dose–response curves for ischemic disease were modelled separately for three age groups [46]. Figure 2 gives an example. As these curves also depend on the frequency of HEDs (see above and [21]), different curves are provided. For people without any history of HED, there are potential beneficial effects at up to 30 g/day average consumption, and these effects are most pronounced in younger ages (see the curve in red versus the curves in green or blue). For people with HED, the curve is flat until it reaches 30 g/day average consumption (black line), with no beneficial or detrimental effects, irrespective of age. After that threshold is reached, the curves are the same irrespective of any history of HED [46].

*Socioeconomic status and wealth (SES)*: Epidemiologic studies using a variety of indicators for SES (education, income, professional status) have consistently shown that, for the general population, morbidity and mortality risk increases as SES decreases [47,48,49]. Behavioural risk factors such as alcohol use and their social patterning have frequently been proposed as factors mediating socioeconomic differences in health [50]. However, there may also be an interaction between such risk factors and SES. Thus, there have been some indications that the dose–response relationships are steeper at lower levels of SES. For instance, the RR of alcohol consumption for those with HIV infections was considerably higher for low SES compared to high SES [51]. In general, systematic reviews and meta-analyses found some indication for an interaction between alcohol use and SES [52]. However, it is not clear if this interaction leading to more harm per litre of ethanol is due to steeper risk curves or due to different drinking patterns or due to differences in reporting (e.g., [53]). The steeper dose–response curves could be due to interactions with other risk factors, such as smoking or BMI (see the discussion on the harm paradox [54]). Overall, exploring this interaction and its possible underlying mechanisms should be a priority in future research endeavours.

The same phenomena can be seen between countries: the harm per litre of alcohol depends strongly on the economic wealth, or on the Human Development Index [55,56]. Analyses stratified by the Human Development Index reveal that a substantial part of the variance for alcohol-attributable all-cause mortality stems from different causes of death between countries. TB once again provides a good example, and has been called the archetypical disease of poverty [57], as it is very much linked to crowding, and other characteristics linked to economically poor environments [58]. Obviously, in rich countries, alcohol consumption and it is effects on the immune system will not lead to TB that often, as there are almost no people with active TB around to contract it from. Thus, the higher risk for the same amount of alcohol can be explained by environmental variables, or interactions with other risk factors, such as crowding. Shield and Rehm [55] provide the list of diseases where various environmental factors play a large role: infectious diseases in general (i.e., all sexually transmitted diseases, including HIV/AIDS, pneumonia) but also liver cirrhosis (via the interaction with hepatitis B and C [59]), and road injuries show the biggest differences in standardized mortality after consuming one litre of pure ethanol.

*Genetic constitution*: Variants of three genes encoding alcohol-metabolizing enzymes, the aldehyde dehydrogenase gene ALDH2, and the two alcohol dehydrogenase genes, ADH1B and ADH1C, have been associated with different risks for some alcohol-attributable diseases. These variants are more prevalent in Japanese, Chinese, and other Asian populations [60]. Differences in the dose–response curves can be found in disease outcomes for which acetaldehyde plays a role (flushing, most alcohol-attributable cancers [61]) and are especially pronounced for oesophageal cancer [62], where acetaldehyde is one of the most important underlying causal pathways [61,63,64].

## 4. Discussion

Before we discuss the findings in detail, we want to point out the limitations. Any review is limited by the quality of its underlying literature. In this case, there are four main limitations: First and foremost, the dose–response relationship for levels of alcohol use depends to a considerable degree on the comparison group. Using the non-drinking group for comparison, i.e., not separating between lifetime abstainers and former drinkers will often lead to more pronounced curvilinear relationships which falsely indicate a beneficial impact at light to moderate levels of drinking for conditions where such benefits do not exist. The reason here is due to the inclusion of the so-called ‘sick quitters’, defined as people who stopped drinking because of health problems [65]. This does not imply that there are no curvilinear relationships between the level of alcohol use and disease and mortality outcomes: as indicated above, there are known biological pathways for the beneficial effects of light to moderate drinking on ischemic disease [36,66,67]. However, the ubiquity of reports on the beneficial health effects from light to moderate drinking [68,69] is mainly due to this effect. Second, most of the risk curves are based on verbal reports of drinkers regarding their consumption levels, thus potentially introducing some biases [44,70,71]. Even seemingly simple questions such as those regarding lifetime abstention may introduce some biases [23]: in a nationally representative US survey with follow-up, more than half (52.9%) of those who reported never having consumed any kind of alcoholic beverage in the 1992 survey had, in fact, reported drinking in previous surveys. Third, most meta-analyses are based on a one-time measurement of alcohol only, with some follow-ups decades later [44]. This assumes that this one-time measurement captures the level of drinking before and after the measurement, and certainly creates regression dilution bias [72] (i.e., underestimation of the true relationship). Finally, in many cases risk curves are based on a few studies from similar cultures. This certainly introduces bias, and also limits the generalizability of our knowledge with respect to groups, defined by sex, age, or other modifiers. Moreover, many of the medical cohorts were selected for their likelihood of returning for follow-up, thus restricting groups with chronic heavy drinking patterns, such as people with alcohol use disorders [73]. For risk curves, this means that the slopes found within the variability of drinking of stable middle-class respondents are simply projected onto slopes for more extreme drinking, where risk acceleration may plateau. This may also create bias, especially for exponentially increasing slopes, and capping the relative risk at values where we have sufficient underlying observations for alcohol exposure may be the answer [74]. Alternatively, RR may be capped at the average risk level for people with alcohol use disorders.

While the above limitations point to the need for more studies to fill in the research gaps, the existing research clearly indicates several implications for alcohol policies, including guidelines: First, as most dose–response curves are monotonous, the lower the level of alcohol consumption overall, the better. While ischemic diseases and diabetes may constitute exceptions, even based on current meta-analyses, it seems clear that less consumption is better (i.e., between 10 and 20 g/day [34,75]), and the risk is sex-specific. This means that most current low-risk drinking guidelines have thresholds which are too high [75]. Second, as many of the dose–response curves are exponential, risk reduction is greater for heavier drinkers compared to moderate drinkers, if both reduce their drinking by the same number of drinks per day [18]. Empirical evidence suggests that this is best achieved by moving the overall population mean downwards [56,76].

## 5. Conclusions

Dose–response relationships are crucial for determining the best medical recommendations (such as low-risk drinking guidelines; [77]) and for creating effective alcohol control policy measures. More research is necessary to better understand their variability and determinants.

## Figures and Tables

**Figure 1 nutrients-13-02652-f001:**
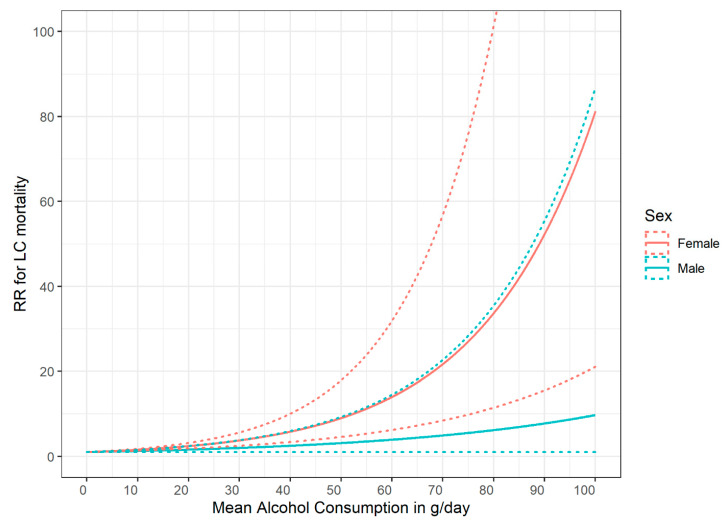
Relative risk for liver cirrhosis mortality as a function of mean alcohol consumption (based on [43]). RR: Relative Risk; LC: liver cirrhosis; solid lines denote point estimates, dashed lines the 95% confidence intervals.

**Figure 2 nutrients-13-02652-f002:**
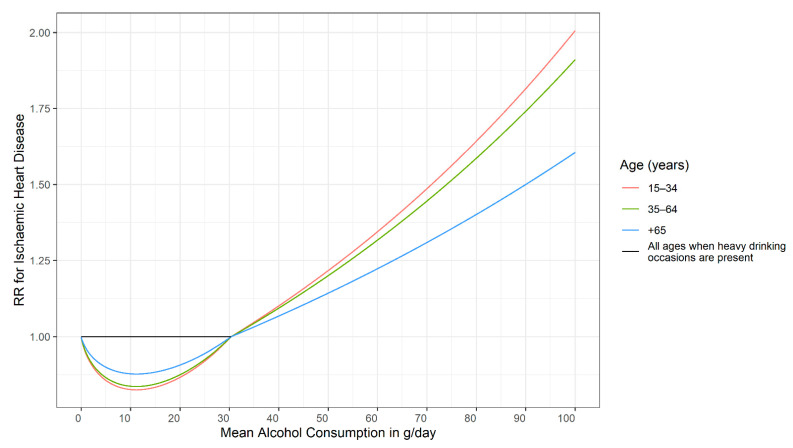
Relative risk for ischemic heart disease mortality as a function of mean alcohol consumption, heavy drinking occasion status, and age for women assuming no heavy drinking occasions for light to moderate drinkers [46]. RR: Relative Risk.

**Table 1 nutrients-13-02652-t001:** Major disease categories causally related to alcohol and modelled in the last comparative risk assessments based on WHO data [8], and their codes in the International Statistical Classification of Diseases and Related Health Problems (ICD).

Global Health Estimate 2015 Cause Category					ICD-10 Coding
I.	Communicable, maternal, perinatal and nutritional conditions				A00–B99, D50–53, D64.9, E00–02, E40–46, E50–64, G00–04, G14, H65–66, J00–22, N70–73, O00–99, P00–96, U04
	A.	Infectious and parasitic diseases			A00–B99, G00–04, G14, N70–73, P37.3, P37.4
		1	Tuberculosis		A15–19, B90
		3	HIV/AIDS		B20–24
	B.	Respiratory infections			H65–66, J00–22, P23, U04
		1	Lower respiratory infections		J09–22, P23, U04
II.	Noncommunicable diseases				C00–97, D00–48, D55–64 (minus D64.9), D65–89, E03–07, E10–34, E65–88, F01–99, G06–98 (minus G14), H00–61, H68–93, I00–99, J30–98, K00–92, L00–98, M00–99, N00–64, N75–98, Q00–99, X41–42, X44, X45, R95
	A.	Malignant neoplasms			C00–97
		1	Mouth and oropharynx cancers		C00–14
			a.	Lip and oral cavity	C00–08
			c.	other pharyngeal cancers	C09–10, C12–14
		2	Oesophagus cancer		C15
		4	Colon and rectum cancers		C18–21
		5	Liver cancer		C22
		9	Breast cancer		C50
		19	Larynx cancer		C32
	C.	Diabetes mellitus			E10–14 (minus E10.2–10.29, E11.2–11.29, E12.2, E13.2–13.29, E14.2)
	E.	Mental and substance use disorders			F04–99, G72.1, Q86.0, X41–42, X44, X45
		4	Alcohol use disorders		F10, G72.1, Q86.0, X45
	F.	Neurological conditions			F01–03, G06–98 (minus G14, G72.1)
		3	Epilepsy		G40–41
	H.	Cardiovascular diseases			I00–99
		2	Hypertensive heart disease		I10–15
		3	Ischemic heart disease		I20–25
		4	Stroke		I60–69
			a.	Ischemic stroke	G45–46.8, I63–63.9, I65–66.9, I67.2–67.848, I69.3–69.4
			b.	Hemorrhagic stroke	I60–62.9, I67.0–67.1, I69.0–69.298
		5	Cardiomyopathy, myocarditis, endocarditis		I30–33, I38, I40, I42
	J.	Digestive diseases			K20–92
		2	Cirrhosis of the liver		K70, K74
		8	Pancreatitis		K85–86
III.	Injuries				V01–Y89 (minus X41–42, X44, X45)
	A.	Unintentional injuries			V01–X40, X43, X46–59, Y40–86, Y88, Y89
		1	Road injury		V01–04, V06, V09–80, V87, V89, V99
		2	Poisonings		X40, X43, X46–48, X49
		3	Falls		W00–19
		4	Fire, heat and hot substances		X00–19
		5	Drowning		W65–74
		6	Exposure to mechanical forces		W20–38, W40–43, W45, W46, W49–52, W75, W76
		8	Other unintentional injuries		Rest of V, W39, W44, W53–64, W77–99, X20–29, X50–59, Y40–86, Y88, Y89
	B.	Intentional injuries			X60–Y09, Y35–36, Y870, Y871
		1	Self-harm		X60–84, Y870
		2	Interpersonal violence		X85–Y09, Y871

## Data Availability

Not applicable.

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
