# Peer review of "Dose–Response Relationships between Levels of Alcohol Use and Risks of Mortality or Disease, for All People, by Age, Sex, and Specific Risk Factors"

_nutrients, 2021, doi:10.3390/nu13082652_

Round 1
Reviewer 1 Report
I enjoyed this review. It’s a useful addition to the literature.
This sentence reads awkwardly “These dose-response curves usually compare the relative change in risk of a certain level of average drinking against the risk of a lifetime abstainer ‘change in risk of’ to ‘change in risk from”. I wonder if the author’s meant “change in risk arising from…” rather than “change in risk of…”
Regarding “The only exception is the Russian Federation and surrounding countries, where region-specific dose-response curves are usually applied [26].” Could the author’s explain briefly why this is?
Regarding Curvilinear dose-response relationships section, although briefly referred to in the discussion, it is a shame to miss an opportunity to make better use of findings from the Wood et al study (2018; 391: 1513–23). Their findings suggest that evidence for protection varies widely among cardiovascular sub-groups and was far more apparent for acute ischaemic events (ie myocardial infarction) than for chronic ischaemic disease (ie non-MI). Very few studies have made this fine level of distinction, yet it is clear that even among ischaemic diseases (ie I20-I25) risk functions vary considerably. If the distribution of acute v chronic ischaemic disease varies between countries, by SES, sex, age and even genetics, then its reasonable to expect that a one-size-fits-all approach to application of RR functions could be more problematic for some populations than others.
Did the authors mean to reference [14] in the following sentences as doesn’t seem a good fit for IHD? “… ischemic heart disease [7, 14?]; ischemic stroke [30]; diabetes [31, 33]”?
Perhaps in the discussion of curvilinear dose-response some recognition of variability by sex is warranted (ie curvilinear finding is stronger and more consistently found for female drinkers for these conditions espc), which could then be explored further in the section on modifiers that follows.
The first two sentences under 3.1.2. are confusing ie it is not apparent why, since males have higher average use and more heavy drinking occasions that it should follow that their health harms should be lower to the same degree? Perhaps a typo here?
Specify time period here: “ For instance, a consumption of 7 and more drinks in women...”
Re “These differences are based on the fact that for dose-response relationships in the select cohorts used in medical epidemiology, there are often not sufficient people included to estimate …” Is that the only reason the curves are steeper for women? Could there be a physiological reason why this is the case for instance?
In the introduction authors make a key point: “People who do not engage in chronic heavy drinking can be separated in two categories: those, who engage in HED and those who do not, with different impacts on some disease and mortality outcomes” and “…this contribution will explore the impact of heavy episodic drinking on dose-response curves”. Given this, would be helpful – if possible -- to include the alternative figure to Figure 2 along side ie RR for IHD for women with episodic heavy drinking occasions but with average light to moderate consumption? Could perhaps free up space by placing Table 1 in supplementary materials.
Regarding “.. not clear if this interaction leading to more harm per litre of ethanol is due to steeper risk curves or different drinking patterns….” Could this observation also be due in part to higher levels of under-reporting among self-reports of consumption by lower SES populations that is related to predominant drinking patterns e.g. recall bias exacerbated for heavy episodic drinkers (ie due to blackouts etc). Or, social drinking practices closely related to economic pressures more common to low SES drinkers? Eg heavy episodic Aboriginal Australian drinkers with lower SES appear to have far greater risk of mortality/morbidity per unit alcohol than general populations. They also have a propensity for purchasing cheap high alcohol content beverages in large containers (e.g. cask wine, fortified wine) that are shared within drinking groups; accurate self-reported use is compromised for heavy episodic drinkers who consume this way.
Author Response
Dear Reviewer 1, thank you for your review. Please see below our point by point response in bold after each question.
Reviewer 1
I enjoyed this review. It’s a useful addition to the literature.
Thank you for this evaluation.
This sentence reads awkwardly “These dose-response curves usually compare the relative change in risk of a certain level of average drinking against the risk of a lifetime abstainer ‘change in risk of’ to ‘change in risk from”. I wonder if the author’s meant “change in risk arising from…” rather than “change in risk of…”
This sentence has been changed.
Regarding “The only exception is the Russian Federation and surrounding countries, where region-specific dose-response curves are usually applied [26].” Could the author’s explain briefly why this is?
We have added an explanation here.
Regarding Curvilinear dose-response relationships section, although briefly referred to in the discussion, it is a shame to miss an opportunity to make better use of findings from the Wood et al study (2018; 391: 1513–23). Their findings suggest that evidence for protection varies widely among cardiovascular sub-groups and was far more apparent for acute ischaemic events (ie myocardial infarction) than for chronic ischaemic disease (ie non-MI). Very few studies have made this fine level of distinction, yet it is clear that even among ischaemic diseases (ie I20-I25) risk functions vary considerably. If the distribution of acute v chronic ischaemic disease varies between countries, by SES, sex, age and even genetics, then its reasonable to expect that a one-size-fits-all approach to application of RR functions could be more problematic for some populations than others.
We have now introduced the variability of ischemic events based on the 2018 paper by Wood et al.
Did the authors mean to reference [14] in the following sentences as doesn’t seem a good fit for IHD? “… ischemic heart disease [7, 14?]; ischemic stroke [30]; diabetes [31, 33]”?
We apologize for the mix-up. We have corrected these references.
Perhaps in the discussion of curvilinear dose-response some recognition of variability by sex is warranted (ie curvilinear finding is stronger and more consistently found for female drinkers for these conditions espc), which could then be explored further in the section on modifiers that follows.
We have added these considerations.
The first two sentences under 3.1.2. are confusing ie it is not apparent why, since males have higher average use and more heavy drinking occasions that it should follow that their health harms should be lower to the same degree? Perhaps a typo here?
It was not a typo, but, since it could lead to a misunderstanding, we have changed it.
Specify time period here: “ For instance, a consumption of 7 and more drinks in women...”
The time period has been added.
Re “These differences are based on the fact that for dose-response relationships in the select cohorts used in medical epidemiology, there are often not sufficient people included to estimate …” Is that the only reason the curves are steeper for women? Could there be a physiological reason why this is the case for instance?
It is not the only reason the curves are steeper for women, and this point is now more clearly stated in the revised text.
In the introduction authors make a key point: “People who do not engage in chronic heavy drinking can be separated in two categories: those, who engage in HED and those who do not, with different impacts on some disease and mortality outcomes” and “…this contribution will explore the impact of heavy episodic drinking on dose-response curves”. Given this, would be helpful – if possible -- to include the alternative figure to Figure 2 along side ie RR for IHD for women with episodic heavy drinking occasions but with average light to moderate consumption? Could perhaps free up space by placing Table 1 in supplementary materials.
We have changed Figure 2 to include this.
Regarding “.. not clear if this interaction leading to more harm per litre of ethanol is due to steeper risk curves or different drinking patterns….” Could this observation also be due in part to higher levels of under-reporting among self-reports of consumption by lower SES populations that is related to predominant drinking patterns e.g. recall bias exacerbated for heavy episodic drinkers (ie due to blackouts etc). Or, social drinking practices closely related to economic pressures more common to low SES drinkers? Eg heavy episodic Aboriginal Australian drinkers with lower SES appear to have far greater risk of mortality/morbidity per unit alcohol than general populations. They also have a propensity for purchasing cheap high alcohol content beverages in large containers (e.g. cask wine, fortified wine) that are shared within drinking groups; accurate self-reported use is compromised for heavy episodic drinkers who consume this way.
This is somewhat speculative, as we need a gold-standard comparison, which usually only exists on a collective level. We have added the thought, however, with a comment regarding the need for future research in this area.
Reviewer 2 Report
This narrative review provides a concise and clear summary of the relative risks associated with alcohol intake, both for amount as a dose-response and for heavy drinking episodes, and effect modification, with evidence drawn from several sources including meta-analyses used by the Global Burden of Diseases Study. The article would benefit from more detail in places, particularly in relation to describing the nature of the associations of alcohol intake with a range of specific diseases and other outcomes.
Some specific comments are below:
Title
- This should be revised to reflect that the article content is not restricted to mortality, but rather risks of disease and injuries.
Abstract
- Line 27-28. Were the RR functions obtained from the 2018 Global Status Report, or from the Global Burden of Disease Study and various other sources, as per methods?
- Line 33: suggest to change wording from “….linked to protective effects….” to “….associated with lower risks compared with not drinking….” as causality has not been established.
- Line 33: suggest to change wording “….alcohol use may interact synergistically….”
- Line 38-39: suggest to change wording “….but certain genetic variants are linked to an increased RR per gram of alcohol consumed for some diseases.”
Introduction
- Line 69-70: The sentence “However, this dose-response relationship will vary based on driving and drinking experience” is a bit unclear in the context.
Methods
- Line 92-93: A more comprehensive description of the sources of evidence used for the review would be helpful.
Results
- Line 136-143: It would be of interest to list and describe some of the major diseases and other outcomes (such as injuries) and the magnitudes of association where a monotonic dose-response relationship with alcohol intake is observed. Similarly, note which diseases and outcomes show exponential or flattening functions and describe these associations and their magnitude. Reference could be made to the range of outcomes in Table 1.
- Line 154-155: Regarding the lower risks of diabetes and ischaemic stroke and heart disease associated with light to moderate drinking, causality has not been established. The potential impact of reverse causation and confounding on these associations, even when the reference group is lifetime abstainers and adjustments for covariates have been made, should be noted,
- Line 159: RR for men missing?
- Line 167: Should this read “…health harms are not higher…”?
- Line 175: It would be of interest to note which diseases other than liver cirrhosis have evidence of higher RRs for women.
- Line 236-239: Change wording to clarify that there is only evidence for certain diseases “different risks of some alcohol-attributable diseases, including xxx”, and also expand on why this effect modification may occur i.e. increased exposure to carcinogenic acetaldehyde for a given alcohol intake among individuals with certain genetic variants.
Discussion
- Line 252: “…there are plausible biological pathways…” the biological pathways relating to potential beneficial effects have yet to be confirmed. Could note what the proposed pathways are.
- Line 253: See earlier comment regarding reverse causation and residual confounding and discuss further here.
- Line 271: Explain what is meant by “capping may be the answer”
References
- Please check that the link works for reference 20.
Author Response
Dear Reviewer 2, thank you for your review. Please find our responses point by point below in bold.
Reviewer 2:
This narrative review provides a concise and clear summary of the relative risks associated with alcohol intake, both for amount as a dose-response and for heavy drinking episodes, and effect modification, with evidence drawn from several sources including meta-analyses used by the Global Burden of Diseases Study. The article would benefit from more detail in places, particularly in relation to describing the nature of the associations of alcohol intake with a range of specific diseases and other outcomes.
Some specific comments are below:
Title
- This should be revised to reflect that the article content is not restricted to mortality, but rather risks of disease and injuries.
We have changed the title accordingly.
Abstract
- Line 27-28. Were the RR functions obtained from the 2018 Global Status Report, or from the Global Burden of Disease Study and various other sources, as per methods?
This has been clarified in the revision.
- Line 33: suggest to change wording from “….linked to protective effects….” to “….associated with lower risks compared with not drinking….” as causality has not been established.
This has been changed as suggested.
- Line 36: suggest to change wording “….alcohol use may interact synergistically….”
This changed to the wording has been made, as suggested.
- Line 38-39: suggest to change wording “….but certain genetic variants are linked to an increased RR per gram of alcohol consumed for some diseases.”
This wording has been changed, as suggested.
Introduction
- Line 69-70: The sentence “However, this dose-response relationship will vary based on driving and drinking experience” is a bit unclear in the context.
We have now clarified this point.
Methods
- Line 92-93: A more comprehensive description of the sources of evidence used for the review would be helpful.
We have added a rationale to the description of the sources.
Results
- Line 136-143: It would be of interest to list and describe some of the major diseases and other outcomes (such as injuries) and the magnitudes of association where a monotonic dose-response relationship with alcohol intake is observed. Similarly, note which diseases and outcomes show exponential or flattening functions and describe these associations and their magnitude. Reference could be made to the range of outcomes in Table 1.
We have added this more in-depth description.
- Line 154-155: Regarding the lower risks of diabetes and ischaemic stroke and heart disease associated with light to moderate drinking, causality has not been established. The potential impact of reverse causation and confounding on these associations, even when the reference group is lifetime abstainers and adjustments for covariates have been made, should be noted,
This caveat had been mentioned, but it is now more highly emphasized.
- Line 159: RR for men missing?
The RR for men has been added.
- Line 167: Should this read “…health harms are not higher…”?
This statement caused misunderstandings for both reviewers so have we changed it.
- Line 175: It would be of interest to note which diseases other than liver cirrhosis have evidence of higher RRs for women.
We have added diseases other than liver cirrhosis which have evidence for higher RRs in women.
- Line 236-239: Change wording to clarify that there is only evidence for certain diseases “different risks of some alcohol-attributable diseases, including xxx”, and also expand on why this effect modification may occur i.e. increased exposure to carcinogenic acetaldehyde for a given alcohol intake among individuals with certain genetic variants.
We have clarified that this phenomenon is relevant only for some diseases, and have introduced reasons for this.
Discussion
- Line 252: “…there are plausible biological pathways…” the biological pathways relating to potential beneficial effects have yet to be confirmed. Could note what the proposed pathways are.
We disagree with respect to whether these pathways have been confirmed, but have now referenced the pathways.
- Line 253: See earlier comment regarding reverse causation and residual confounding and discuss further here.
Please see response to #14 above.
- Line 271: Explain what is meant by “capping may be the answer”
We have added a further explanation on what we mean by capping.
References
- Please check that the link works for reference 20.
Yes, we checked and the link for reference 20 works.